# Is scientific evidence enough? Using expert opinion to fill gaps in data in antimicrobial resistance research

**Melanie Cousins**[1]*, **E. Jane Parmley**[2], **Amy L. Greer**[2], **Elena Neiterman**[1], **Irene A. Lambraki**[1], **Tiscar Graells**[3,4], **Anaïs Léger**[5], **Patrik J. G. Henriksson**[4,6,7], **Max Troell**[4,6], **Didier Wernli**[5], **Peter Søgaard Jørgensen**[3,4], **Carolee A. Carson**[8], **Shannon E. Majowicz**[1]

**1** School of Public Health Sciences, University of Waterloo, Waterloo, Ontario, Canada, **2** Department of Population Medicine, Ontario Veterinary College, University of Guelph, Guelph, Ontario, Canada, **3** Global Economic Dynamics and the Biosphere, Royal Swedish Academy of Sciences, Stockholm, Sweden, **4** Stockholm Resilience Centre, Stockholm University, Stockholm, Sweden, **5** Global Studies Institute, University of Geneva, Geneva, Switzerland, **6** Beijer Institute of Ecological Economics, Royal Swedish Academy of Sciences, Stockholm, Sweden, **7** WorldFish, Penang, Malaysia, **8** Foodborne Disease and Antimicrobial Resistance Surveillance Division, Centre for Foodborne, Environmental and Zoonotic Infectious Diseases, Public Health Agency of Canada, Guelph, Ontario, Canada

* melaniemcousins@gmail.com

**Data Availability Statement:** The data is held in a public repository Boreal at the following citation: Cousins M. Using expert knowledge and experience to parameterize a simulation model of

## Abstract

### Background

Antimicrobial Resistance (AMR) is a global problem with large health and economic consequences. Current gaps in quantitative data are a major limitation for creating models intended to simulate the drivers of AMR. As an intermediate step, expert knowledge and opinion could be utilized to fill gaps in knowledge for areas of the system where quantitative data does not yet exist or are hard to quantify. Therefore, the objective of this study was to identify quantifiable data about the current state of the factors that drive AMR and the strengths and directions of relationships between the factors from statements made by a group of experts from the One Health system that drives AMR development and transmission in a European context.

### Methods

This study builds upon previous work that developed a causal loop diagram of AMR using input from two workshops conducted in 2019 in Sweden with experts within the European food system context. A secondary analysis of the workshop transcripts was conducted to identify semi-quantitative data to parameterize drivers in a model of AMR.

### Main findings

Participants spoke about AMR by combining their personal experiences with professional expertise within their fields. The analysis of participants' statements provided semi-quantitative data that can help inform a future of AMR emergence and transmission based on a causal loop diagram of AMR in a Swedish One Health system context.

AMR emergence and transmission in a Swedish food system context: Framework Matrices [Internet]. Borealis, V1. 2022. p. UNF:6:7MNV1+/sxDeftXfWjznwkg== [fileUNF]. Available from: https://doi.org/10.5683/SP3/WUXL5F.

**Funding:** This study is funded through an operating grant of the 5th Joint Programming Initiative on Antimicrobial Resistance (JPIAMR 2017). Funding was provided by an operating grant from the Canadian Institutes for Health Research (Institute of Infection and Immunity, Institute of Population and Public Health: https://cihr-irsc.gc.ca/e/193.html) (PI: SEM, grant number 155210); a Swedish Research Council grant (https://www.vr.se/english.html) (PI and project consortium coordinator: PSJ, grant number 2017-05981); and an operating grant from the Swiss National Science Foundation (https://www.snf.ch/en) (PI: DW, grant number 40AR40_180189). The funders had no role in the study design, data collection and analysis, decision to publish, or preparation of the manuscript.

**Competing interests:** I have read the journal's policy and the authors of this manuscript have the following competing interests: EN, TG, and DW have no declarations of interest to report. MC and IAL work for the Public Health Agency of Canada. EJP is engaged in research funded by the Canadian Institutes for Health Research, Natural Sciences and Engineering Research Council, Ontario Ministry of Agriculture, Food and Rural Affairs, the Public Health Agency of Canada, and the Canadian Safety and Security Program. She is currently President of the Board of Directors of the Centre for Coastal Health, president of the Canadian Association of Veterinary Epidemiology and Preventive Medicine, member of the Board of Directors of the McEachran Institute, member of the Advisory Council for Research Directions: One Health, and a member of the Royal Society of Canada One Health Working Group. Prior to February 2019, she was employed by the Public Health Agency of Canada. ALG is engaged in research funded by the Canadian Institutes for Health Research, Natural Sciences and Engineering Research Council, Agriculture Canada, and Canada First Research Excellence Fund. She has served as a consultant for the Ontario Secondary School Teachers Federation (OSSTF) and as an expert witness in legal proceedings related to the SARS-CoV-2 pandemic. She is currently a member of the Board of Directors for the NSERC Emerging Infectious Disease Modelling Network – OMNI network, and Advisory Board Member of the National Collaborating Centre for Infectious Diseases (NCCID). Prior to January 2014, she was

## Conclusion

Using transcripts of a workshop including participants with diverse expertise across the system that drives AMR, we gained invaluable insight into the past, current, and potential future states of the major drivers of AMR, particularly where quantitative data are lacking.

## Introduction

Antimicrobial resistance (AMR) is one of the most devastating global problems [1–4] that led to 4.95 million deaths in 2019 [5] and has had other devastating impacts on the health and well-being of humans, animals, and the environment (e.g., increased hospital stay and death, loss of production [1–4]). The major driver of AMR has been commonly reported to be the use of antimicrobials (AMs) [1–4]. However, there are many more distal drivers of AMR that relate to why and how we use AMs, which stem from cultural, social, and economic conditions [1–4, 6–9]. Due to the complexity of AMR and the intricate social and economic dynamics that underpin much of the system of drivers of AMR, AMR has yet to be discussed and dealt with at a broad scale [7–10]. Many interventions to combat AMR are siloed to single sectors and, if implemented, may have unintended consequences in the broader system, may not be adopted into policy, or those adopted may be met with non-compliance [3, 9]. For example, many policies and regulations that try and limit antimicrobial use (AMU) have failed to account for some of the underlying reasons for use (e.g., overcrowding and unsanitary conditions) and therefore are not easily adopted by those they are intended to regulate (e.g. producers continue to use AMs as a cheaper alternative to making costly investments in their facilities [9, 11, 12]), or may lead to unintended consequences (e.g., purchasing of AMs on the black market, the loss of animal lives and production, the shift to more potent, critical, or broad spectrum antimicrobials [9, 13]). Therefore, to better address the issue of AMR, it is necessary to understand this problem from a systems view and engage stakeholders in exploring the *why* and *how* of the issue to be able to identify drivers of AMR and how they may influence each other.

One way to better understand how a system works is through quantitative, qualitative, or mixed methods simulation models of said system [14–20]; a representation of the operations of a real-world process or system over time [15]. Simulation models have been used to explain and predict the emergence and transmission of AMR; however, these models are rarely integrated across sectors and usually focus on small populations in specific settings [21]. Creating complex and integrated models that capture the diverse One Health aspects of AMR requires a large amount of data. Literature reviews on models of AMR have identified that one of the major limitations to the creation of integrated models of AMR is the limited empirical data to parameterize the complex models and the lack of measures for how the various parts of the system impact each other [21–23]. Therefore, to further modelling of AMR and to better assess interventions for combating this important global issue, other types of knowledge, such as professional knowledge, experiences, and opinions (tacit knowledge), may be able to address existing knowledge gaps across the broader system.

A participatory modelling workshop, consisting of experts from the One Health system in Europe outlined the system that drives AMR and co-created a causal loop diagram (CLD; a visual representation of all of the key variables (called factors or nodes) and all of their interconnections (called relationships) within the system; [24]) of the development and transmission of AMR within a European One Health system context [9]. The result of this workshops

employed by the Public Health Agency of Canada. PJGH is partially funded by FORMAS Inequality and the Biosphere Project (2020-00454) and partially by CGIAR Trust Fund. PJGH is a member of the Technical Committee for the BAP Vanguard Climate Action Standard and PJGH and MT act as scientific advisors to the Seafood Business for Ocean Stewardship (SeaBOS) initiative. PJGH was supported by the CGIAR Research Program on Fish Agri-Food Systems (FISH) led by WorldFish and on Climate Change, Agriculture and Food Security (CCAFS) supported by contributors to the CGIAR Trust Fund. PJGH and MT acknowledge the Kjell and Märta Beijer Foundation for supporting this work through the Beijer Institute's Aquaculture and Sustainable Seafood programme, and the SEAWIN project funded by FORMAS (2016-00227). PJGH is partially funded by FORMAS Inequality and the Biosphere project (2020-00454). AL works for the Swiss Federal Food Safety and Veterinary Office. PSJ was also funded via an ERC starting grant: INFLUX, grant number 101039376. He holds/has held grants as principal or co-investigator, from the following agencies and foundations all related to the topics of social-ecological systems and/or AMR: Swedish Research Council FORMAS, Wallenberg Foundations, IKEA Foundation, Erling Persson Family Foundation. CAC works for Government of Canada at the Public Health Agency of Canada. In that role she has been a subject matter expert for the World Organisation for Animal Health (WOAH) and the World Health Organization (WHO). She is a member of the advisory committee for Animal Health Canada. She has previously been engaged in research funded by the Canadian Institutes for Health Research, Ontario Ministry of Agriculture, Food and Rural Affairs, and the Canadian Safety and Security Program and has been a co-topic editor for two volumes of a Frontiers in Veterinary Science, Research Topic: Antimicrobial Usage in Companion and Food Animals: Methods, Surveys and Relationships with Antimicrobial Resistance in Animals and Humans. SEM is (or has been in the last 5 years) engaged in research grants/contracts funded by the Canadian Institutes for Health Research, the Bill and Melinda Gates Foundation/ UK Dept., International Development, the Natural Sciences and Engineering Research Council of Canada, Institut de recherche Robert-Sauvé en santé et en sécurité du travail, Dairy Farmers of Canada Research Funding Program, the World Health Organization, British Columbia Centre for Disease Control, Indian and Northern Affairs Canada, and Ontario Ministry of Agriculture, Food and Rural Affairs. She currently sits on the Editorial Boards of Foodborne Pathogens and Disease, and

was a qualitative description of the system, however, during the conversations about AMR, many quantifiable statements were made regarding various areas of the system. Therefore, the goal of this study was to derive quantifiable data from statements made by a group of experts from the One Health system that drives AMR development and transmission in a European context (further referred to as the system). Specifically, we were interested in the content of statements made by experts in terms of the objective indicators they reported (e.g., the current state of the main factors and the strength and direction of relationships between drivers), and the evidence they used to support their statements (e.g., tacit, or explicit knowledge).

## Methods

This paper expands on a previously published study by Lambraki et al., 2022, which identified AMR drivers and their interconnections in a European One Health system [25] context, through two participatory modelling workshops [9]. While the original study outlined in Lambraki et al., 2022 produced a CLD of the system that drives AMR and thematically described the major areas of the system and potential places to intervene, it did not quantify the current state of the factors or put strengths to the relationships between the factors. Therefore, the transcripts from the workshop were re-analyzed to begin to quantify the CLD by identifying quantifiable data from the experts' statements for the factors and strengths and directions of the relationships between the factors. Full details of the workshop methods and outputs are provided in Lambraki et al. [9], but here we provide a brief overview of the workshop setup, the participants involved, and the major outcomes of the workshops relevant to the secondary analysis we conducted for this study [9]. The study received ethics clearance from the University of Waterloo's Research Ethics Committee (ORE# 40519) and all participants gave written consent to participate.

### The workshops and participants

Two model-building workshops, each lasting about 6.5 hours, took place on September 19[th] and 20[th], 2019 at the Stockholm Resilience Centre in Stockholm, Sweden. The goal of participant selection was to identify individuals that have knowledge about AMR and those that may not have knowledge about AMR but work in sectors/areas that may impact or be impacted by the problem. To this end, a matrix of perspectives from sectors that may directly or indirectly impact AMR across the One Health spectrum was developed. The matrix was populated with names and contact information of people that fit the desired perspectives via Google, LinkedIn, and Twitter searchers, organizational websites, and research team networks. Sixty-four participants were approached via email with a maximum of two follow-up contacts in alignment with the University of Waterloo Ethics Committee approved protocols. When an individual representing a particular perspective declined or did not respond to our invitation email and follow-ups, we approached another individual on our complied list representing the same perspective. Overall, 26 participants did not respond, twenty-one declined due to work conflicts or because AMR was not relevant to their work or area of expertise.

In total, seventeen participants participated, which allowed for the collection of rich information and information redundancy [9, 26–29]. Workshop participants possessed varying levels of knowledge about AMR (from no background or understanding of AMR to high level of knowledge). Perspectives represented at the workshop included: epidemiology, food safety and microbiology, veterinary sciences, aquatic sciences and aquatic foods, agricultural crops and policy, animal welfare, human medicine, nursing, public health, public health advocacy, consumer advocacy, pharmaceutical marketing, pharmaceutical law, trade and economics, urban agricultural innovation, sustainable foods and innovation, dietetics, peace and conflict

Epidemiology and Infection, and is a member of the World Health Organization's Foodborne Disease Burden Epidemiology Reference Group, and she has served as a paid expert on behalf of the Attorney General of Canada in legal proceedings, providing evidence on the public health risks and benefits of unpasteurized milk. This does not alter our adherence to PLOS ONE policies on sharing data and materials.

resolution, and leadership. These participants represented organizations at sub-national, national, or regional (i.e., European) levels, including governmental and non-governmental organizations, health care organizations, private consultants, and industry. Over half of the participants were from Sweden, and the remaining from France, Italy, Spain, United Kingdom, and Belgium. Participants did not have previous relationships with research team members. Most of the participants had experience working with organizations across the European Union and Europe at large and internationally.

The two days were identical in structure and intended outcomes, however, they differed by the types of participants involved. The first day consisted largely of "AMR experts" who had expertise in AMR within various areas of the overall food system (e.g., nursing, veterinary medicine, food safety). These experts were engaged first to give a better understanding of the state of AMR within Europe broadly. The second day was mainly made up of participants who were considered "non-traditional experts", who were individuals with expertise within the broader system, but are not traditionally engaged in discussions of AMR (e.g., consumer advocacy, animal welfare, pharmaceutical law). Using open-ended questions and group discussion, the participants and the facilitators physically mapped out the major drivers of AMR (nodes) and sought to identify how these drivers were connected (relationships). Participants were asked to describe the nodes in measurable terms (i.e., something that can increase or decrease) versus more subjective descriptions. To begin, participants were asked where they felt their expertise "fit" into a CLD of AMR, originally made for the Canadian context [24], and if there were any aspects missing or that needed to be changed or removed from the CLD, to reflect the European context. The final CLD is available online [9]. The workshops produced rich volumes of text containing descriptions about factors and how they impact other factors resulting in a thematic description of the system and a CLD of the European food system that drives AMR.

## Data analysis and approach

For this study, the transcripts of the above workshops were coded in NVIVO 12 using the same codebook from the previous study [9], as well as allowing for open coding to identify new, missed, or refined themes. The codebook was originally created for the purpose of identifying major drivers of AMR and relationships between the drivers [9]. For this study, similar to a previously conducted participatory study aimed at quantifying drivers of resistance [25], additional codes were added to identify the level of the nodes (high, medium, low, none, unknown, or varies throughout Europe). The level of the node refers to position of that node in Sweden on a scale of the amount, quantity, extent, or quality compared to a referent (e.g., Sweden versus other countries within or outside of Europe, Sweden currently versus historically). For example, there is lower AMU in agriculture in Sweden now compared to ten years ago [30] or there are low levels of AMR in Sweden compared to low- and middle-income countries [31]. Codes were also used to identify the strength (strong, weak, not mentioned) and direction (positive, negative, not mentioned) of the correlation of relationships between the nodes.

In their accounts, participants referred to a variety of sources of data; this was of particular interest because it helped to identify the areas where scientific evidence exists or is absent. The tacit knowledge and practical experiences shared by the experts could help to inform a model, by filling the gaps in the published evidence and validating evidence from the literature. Some of these accounts were explicitly stated, in which the experts stated that data exists or does not exist, referred to the specific government, scientific reports, or studies for which the data they were referring to, or described an experience from their work (e.g., "*when I was a nurse. . .*", "*at our company we. . .*", "*in my professional opinion. . .*" were all examples of accounts related to professional experience or opinion). Other times the source of the data was implicit and was

revealed through the language used (e.g., "*it is well known that. . .*" implies general knowledge). Therefore, additional codes were added to capture the source of the data related to AMR: general knowledge; personal opinion and experience; professional opinion or experience; and scientific evidence. General knowledge was used for knowledge that the general public or the "lay person" would know from encounters through their daily lives as opposed to knowledge on a specific subject that would result from training or exposure to a specific area. Scientific evidence was further broken into three levels based on perceived quality or amount of data that exists for a given node: low–little or no data currently exists (e.g., surveillance or research has yet to be done), medium–poor, inconsistent data, proxy data used; high–good data, experimental studies, published literature or surveillance reports.

To ensure the coding was consistent, intercoder reliability [25] was assessed between three independent researchers on 10% of the nodes (n = 12) and relationships (n = 20). There was 61% consistency between the coders, which reflects the subjective nature of the coding. Inconsistencies were minor, with the major source of discrepancies due to misunderstanding on the different sources of knowledge (mainly personal versus professional). Discussions occurred between reviewers to better understand the definitions (which were refined for clarity) and came to 100% consensus about our discrepancies, which was our indicator that satisfactory reliability had been reached [32–36].

Framework analysis [37–39] was used to organize the codes into a matrix showing the intersection between the node of interest, the level associated with that node (Table 1), and the source of data to support the claim (Table 2). The framework matrix was generated in NVIVO 12 and then exported to Excel to be organized and refined. A separate matrix was created for each workshop, refined through discussion with the qualitative expert on the research team (EN) until consensus was reached, and then combined to represent a collective view of the participants regarding the system.

During the analysis, it became apparent that a framework matrix could not fit the data related to the relationships due to the complexity and number of interconnections identified from the transcripts. Therefore, a concept map [40–42] was created to visually represent these connections with colour and weighted lines depicting the strength of the relationships and the type of evidence used (explicit or tacit knowledge) respectively, and symbols (+/-) depicting the direction of the correlation of the relationships. The concept map was designed using miMind version 3.13 (Fig 1).

Nodes were grouped under larger headings based on the coding scheme, depicted through large bubbles representing the broader concept (e.g., AMR) and smaller bubbles inside representing the more specific aspects of the concept (e.g., AMR in humans, AMR in food-producing animals). Relationships could be between the broad concepts or the specific nodes depending on the level of detail provided by the participant. The nodes (which were visually represented as a box) were then colour coded and shaded to reflect the level (colour) and the source of the data (darkness of the shading) which was informed by the framework matrix. When two or more claims were made pertaining to the same node or relationship with varying levels of detail, the statement that was specific to Sweden was chosen for visual representation. Similarly, when two or more statements were made about a node or relationship using different sources of evidence, the following hierarchy was used to determine which was used to visually represent the evidence: scientific evidence > professional > personal > general knowledge. In instances where participants had conflicting views in relation to the level of the nodes or the strength or directions of relationships, the opinion of the majority of participants was used on the concept map, and both views were noted in the framework matrix.

The combined concept map created from the two workshops was compared to the existing CLD [9] to ensure that the nodes and relationships identified in this analysis appeared and

**Table 1. Sample combined framework matrix with quotations showing how workshop participants explained the level at which four different drivers of AMR in the Swedish One Health system context exist, stratified by expert type (1: Traditional AMR experts, workshop day 1; 2: Other experts in upstream drivers of AMR, workshop day 2).**

| Level | Access to AMs outside the system | Agreements, regulations, and standards: compliance and enforcement | Non-AM disease prevention | Antimicrobial Resistance |
|---|---|---|---|---|
| High | | *"I just want to add, just a small thing about the legislation, and it was a project some years ago called Eco Welfare, where they looked at different countries, and how they implemented the legislation and so on, and Sweden were, we are relying a lot on the legislation, and we are really, we are following the legislation"* (2) | | |
| Low | | | | *"Oh that is what I started with all my lectures in every country where I go. I go to a lot of countries and tell them about Sweden, and Sweden is one of the extreme positive examples of the world. . . we had today an extremely good situation when it comes to resistance. . ."* (2) |
| None | | | *"We are very rarely applying a lot of the preventative measures we know we could, regardless whether that is changing our role, our behaviours, strong vaccination and stuff like that, and vaccination in one place."* (2) | |
| Unknown | *"Of course, we don't take into account black market operations or internet sales and stuff like that which are tricky"* (1) | | | |
| Varies in Europe | | | | *". . . there are other such maps mapping the situation globally and in Europe, and it is obvious that we are living in a country with extremely privileged situations when it comes to resistant, resistance. Together with, I should say, the other Nordic countries and the Netherlands, which is the good example to show that it is not only a north, south effect of Europe because generally [names of countries] fares a lot worse than we do. . ."* (2) |

matched those in the CLD, which was previously validated with workshop participants through member checking [9, 28, 43]. Nodes and relationships that were not found in the original CLD (nodes: n = 35, relationships: n = 74) were noted for further discussion with the broader research team for inclusion in the final model.

## Findings

### Framework matrix of nodes

There was a total of 83 nodes included in the framework matrix: 48 nodes were found within the original CLD (n = 40) or its overarching factors (n = 8) [9], and 35 were new nodes that were created and added to the framework matrix from this analysis. These 35 nodes emerged

**Table 2. Sample combined framework matrix with quotations showing the source of the data workshop participants were assumed to have been used when describing five different drivers of AMR in the Swedish One Health system context, during conversations with participants (P), facilitators (F), and other research team members (R), stratified by expert type (1: Traditional AMR experts, workshop day 1; 2: Other experts in upstream drivers of AMR, workshop day 2).**

| Source of data (data amount or quality) | Access to AMs outside the system | Appropriate prescribing, diagnosing, treatment: Prescription necessary for AMs | Burden of illness: Human illness | Resistance: Resistance in wider environment | Urbanization and population growth |
|---|---|---|---|---|---|
| **Scientific evidence (High)** | | P: *"Just on the regulatory side we talked about a few minutes ago here in the EU, I understand all antibiotics for humans and animals are by a prescription by a medical doctor, veterinary doctor or veterinary surgeon. So it's the professional vets and professional doctors who have to give a prescription for use."* <br> P: *"Even though in Europe it is a little bit different because everything leads to a prescription".* | | | |
| **Scientific evidence (Medium)** | | | *"It also falls on the human side of course, but just as well as when we talk to microbiologists about our surveillance systems for antimicrobial systems, and if some microbiologists as soon as they realize that the samples may not be taken the same way in each hospital or the cut off, for when you take a blood sample it is not the same. The immediately say, it cannot be used. You cannot compare this data. And every time we have to say, well this is the best data we have. . ."* (1) | | |
| **Scientific evidence (Low)** | | | | *"I mean you know we know actually nothing about really what is going on in the natural environment, largely because much of that research is just not been funded. . ."* (1) | |
| **Professional experience/ knowledge** | *"And of course not everyone is buying antibiotics to begin with but people pass them along the family to friends, and some people get them abroad when travelling because it's easier than in the country that they live in. So it's the whole mobility aspect as well."* (1) | P: *"I think. . .one question is also for instance in Sweden and I think also Europe nowadays recently, you cannot buy and just going into a store, but I know in many other countries you can buy antibiotics yourself."* <br> R: *"Yea."* <br> P: *"You do not even have to have a prescription. So, I think that is a very, very important."* (2) | | | |

*(Continued)*

**Table 2.** (Continued)

| Source of data (data amount or quality) | Access to AMs outside the system | Appropriate prescribing, diagnosing, treatment: Prescription necessary for AMs | Burden of illness: Human illness | Resistance: Resistance in wider environment | Urbanization and population growth |
|---|---|---|---|---|---|
| Personal experience/ knowledge | | | *"So we are less prone to suffer from such infections I think than. . . than malnourished in African, I mean if you take, you take that as a great example, and we are also more prone to go to the doctor immediately in these cases, which is a problematic thing, we are really healthier."* (2) | | |
| General Knowledge | | | | | *"What about the increased connectivity among people globally. The concentration of people, urbanization. I mean all those kind of very large factor"* (1) |

from: 1) new nuances that came to light during this analysis, or 2) nodes broken down into subcategories or merged into broader categories (e.g., AMR as a broad category; AMR in humans, food-producing animals). The latter was important because sometimes the broad category was referenced instead of the specific node. For example, one conversation that took place mentioned infection prevention and control measures in broad terms, "*we very rarely applying a lot of the preventative measures we know we could.*" This excerpt was part of a discussion on how we as a society are not doing enough in terms of prevention measures. However, some participants referred to a specific sector, for example, "*these countries in some of the hospitals, they don't have any infection control nurses or any infection control staff at all.*" This claim was directly related to infection prevention and control measures within the healthcare system (specifically in hospitals).

There was a broad range of topics covered in the two workshops that spanned many sectors (e.g., humans, animals, environment) and scales (e.g., sub-national, national, international). Excerpts from the combined framework matrix are depicted in Tables 1 and 2 (full framework matrix [44], which shows the variety of topics (nodes) covered, tabulated against the associated level of the node (Table 1) and the source of the data that was either explicitly stated or was implied through the participant's wording (Table 2).

Although transcripts were coded using "high", "medium", and "low" codes, statements were only made in language that referenced "high" or "low" but not in the "medium" category and thus it was dropped from the finalized framework matrix. A total of 27 nodes were categorized as "high", 23 as "low", 23 as "unknown", 8 as "none", and 16 nodes were said to vary across Europe.

Strong language was used to refer to "high" levels, such as in the case of one participant who said that Sweden "*. . .is a huge importer of chicken meat, beef, even pork from [name of country], which are produced under completely different conditions concerning the environment, concerning the use of antimicrobials. . .*" This language implied a high or even very high level of importation by Sweden. This participant continues to discuss how this was of concern because some imports could come from countries that may condone some unfavourable agricultural practices and increase Sweden's exposure to AMR and AM residues.

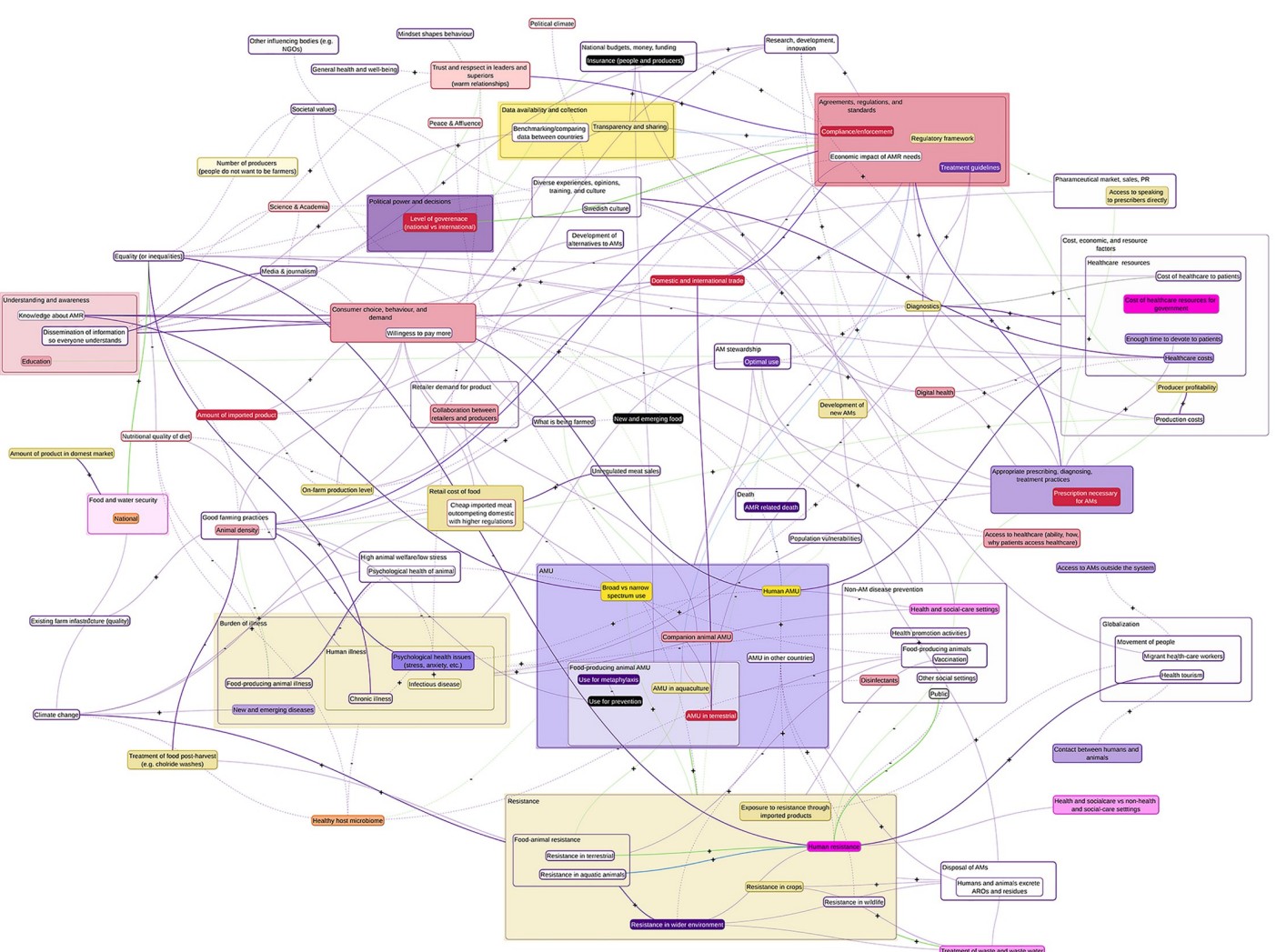

**Fig 1. Combined concept map from the two workshops in which participants described the drivers of antimicrobial resistance in a Swedish One Health system context.** The map consists of nodes (bubbles) and relationships (arrows) referenced by the participants in which the colour of the nodes represents the level at which the drivers of antimicrobial resistance exist, the amount of shading of the nodes represents the source of the data, the colour of the arrows represents the strength of the correlation of the relationship, the weight of the arrow represents the source of the data, and the direction is represented by +/- when available. Note: this figure is zoomable in the PDF version to legible font size.

Alternatively, strong language was also used to refer to "low" levels, such as when one participant said "... *actually during 2019 WHO [World Health Organization] has tried to boil down all the resistance is to all bacteria into one score, to simplify it, and then Sweden comes out on top, [name of country] comes out in the bottom*". In this case, the participant was referring to Sweden as having low levels of AMR in general compared to other countries.

There were a few instances (8 out of the 83 nodes) where Europe (specifically Sweden) was mentioned to have "zero" or "none" for a given category. For example, quota for meat, dairy, and eggs was identified to not be an important part of the agricultural system in Sweden (it is an "*absolutely free market*"). Also, AMU for growth promotion and (soon to be) for prevention of disease under certain conditions is banned in the European Union (EU), post-harvest interventions for disease control (e.g., chloride washes) are not common practice, nor is insurance

for producers who may lose crops or animals to disease, and finally selling insects for human consumption is illegal (see framework matrix for quotations pertaining to these nodes [44]).

In terms of the evidence used by participants, the majority of statements (52 of 83 nodes) were coded as professional experience and opinion (tacit knowledge) based on observations in the participant's professional background. For example, one participant in the second workshop mentioned:

> . . . *Since I have worked with that for twenty years. . . I can see what have happened in intensive care, and I think it is, it is generally in Sweden because before, people, people, nurses and doctors used their craft, the hand craft. They.., they exam[ine]. . . patients much more. Today it's, the reference is the computer system. You take a long [inaudible] report. You read and then you have the reference about, you know, this patient, and then we got into it, and see, oh, it is not as it was written, or I was reading in there.*

This quotation suggested that there has been a change in healthcare practice in Sweden and its potential impact on healthcare professionals' ability to diagnose patients. Based on personal recollection, this opinion was categorized as tacit knowledge.

There were eight claims that were categorized as referencing participants' personal knowledge. For example, one participant mentioned:

> *I don't know the data about Sweden, but I have this guesstimate based on someone that knows. . . how are people feeling in Sweden, like, in terms of the stress level, depression, psychological wellbeing, because I know my own experience as a young kid and older kid back in the day, I was incredibly stressed, focused on like producing stuff and not getting well, and then I didn't care what I ate. . . So I am wondering how are the systems that we live in affecting like long term preventions perspective. Our stress levels, and how is that affecting us.*

This participant provided examples from their personal experience to generalize about the state of psychological health and stress levels in Sweden and how it relates to the current culture and pressure to perform.

There were four statements coded in the category of general knowledge. One example was from the second workshop in which one participant mentioned, "*. . .we have no wars in Sweden for two hundred years, at least for two hundred years*" In this case, the participant was stating that Sweden has been in a state of peace for many years (thus the node Peace and Affluence was categorized as "high"). This statement was made in the context of AMR, stating that because Sweden has been at peace, they have had more time and resources to devote to combatting long-term issues such as AMR. This was in contrast to some countries which may be engaged in war or dealing with other more time-sensitive matters such as political unrest and international conflicts.

The last category of evidence are those statements that refer to the scientific evidence (or lack of evidence) for a given node (n = 33). Some participants specifically referred to there being no data or that the existing data are of very poor quality, which was coded as "scientific evidence–low" (n = 7). These types of statements were usually used to highlight the lack of data and the need for better data in these areas as illustrated in this representative example:

> *Participant*: I mean antimicrobial use sounds easy. . . we can get sales figures. But, split packs and things like this, this gets chucked away and doesn't get chucked away.
>
> *Participant*: I mean we are not even measuring it.

*Participant*: *We don't actually have precise figures on use. Most of the figures used are based on sales from pharmaceutical companies, or from prescription figures from definitely surgeons, or doctors and so on and they are very broad aggregate figures. How many of those are actually used, we really don't know. We just assume that the sales figures are a good proxy...*

This conversation highlights that the way in which AMU in human medicine and agriculture is measured currently is not an adequate or reliable measure and that there are many reasons for this.

The claims that referenced scientific or experimental data or data that was were referred to as more accurate ("scientific evidence–high") occurred for 13 of the 83 nodes. An example of a claim backed up by good scientific evidence from the second workshop (Table 2) which referenced a study that was performed, the name of the study, and the results from that study to back up their claim that Swedes are "rule-followers" and tend to adhere to regulations and legislations in general.

## Map of drivers and relationships

Overall, there were 189 relationships mentioned, and a direction for the correlation of the relationship was either explicitly stated or was easy to decipher from the example using the team's background knowledge of the AMR system (131/189 directions deciphered). However, the strength of the relationship was less commonly reported or able to be deciphered from the language used during discussions (32/189 strengths deciphered). In this case, the "unknown" category commonly represented a claim that did not contain language that would indicate the strength of relationship (see purple lines in Fig 1). For example, one participant in the first workshop mentioned "*this [research and development] will lead to better gathering of data, sharing of data, which will in turn lead to better prioritization of policies and also allow us more budget around the whole system and within the system for each species and I think it is all this systematic approach and it will take a lot of time.*" This quotation mentioned a lot of relationships that are important to understand how the research and data drive each other, and how that leads to policy and opens up budgets for further research. This participant gave insight into the direction of these relationships through the language they used (x will lead to better y is indicative of a positive relationship). However, they did not use language to indicate the strength of the relationships (e.g., x will lead to a lot/a little better y). This quotation was coded as professional opinion as they referred to budgeting as "us" and therefore positioned themselves professionally within the context.

Overall, most relationships were mentioned without an indication of the strength of the relationship (n = 157/189), 28 indicated strong relationships, and very few indicated weak relationships (n = 4). Two of the weak relationship claims were made in comparison to their strong counterpart. For example, participants mentioned that the relationship between AMU in terrestrial food-producing animals and the risk of AMR in humans had a strong relationship, compared to the use of antimicrobials in aquaculture, which participants described as posing less risk (or a weak relationship) of AMR development in humans. The other weak relationship claims made by participants were categorized based on the language used in the claims that indicated the relationship did not really exist or was not overly important in the European context. For example, for the Sweden context, one participant mentioned that "*we don't see increased deaths in untreated [illness] for example, or we don't see that children mortality is going up even though we have reduced the antibiotic use enormously...*" This indicated that there was a weak association between AMU and deaths in humans in Sweden, whereas this may not be the case in other countries (e.g., low- and middle-income countries where untreated infections may more often lead to death).

There were many instances in which personal (n = 80/189) and professional opinion and experience (n = 95/189) were used to back up claims of relationships between nodes. Five of the relationships were supported by statements which we categorized as general knowledge and scientific evidence was used to support data to inform 24 of the relationships, of which only 5 indicated the strength (all categorized as strong) of the relationships. For example, when discussing how consumers can have a large influence on the government, one participant mentioned:

> *Some of those triggers can be for example, the, the making the transparency, increasing transparency, making data by the book [available] to general members of the society, and so that they are aware of what a situation is, . . . I am guessing about what is happening in the Netherlands with the* ESD *[Environmental Systems Division] . . .. That is what triggered the decision of the Minister to say, 'okay, now we will implement targets [for] use and I want to see this done by a year two or year three, and I want 75% reduction in the use of antimicrobial In farm production.' . . . That was all driven by newspapers showing [the] data.*

This participant provided a specific example and based on their understanding of the situation described how an increase in data transparency and making data more available to the general public (e.g., through news and media) led to a change in consumer demand for products (e.g., a reduction of AMs used for food agriculture), which in turn led to large change in government decisions (implementation of targets for agricultural AMU to reduce by 75%) and caused a large reduction in AMU. This participant's claim gave insight into the strength and direction of the relationship and referenced a scientifically based indicator.

Interestingly, sometimes the statements made by one participant in reference to the strength and/or direction were followed by another participant who provided additional evidence from their own personal experience or professional knowledge (or vice versa) to collectively create an evidence-based statement for the relationship. In one example, one participant mentioned "*. . .you would improve your farming practices, and therefore in the short term there would be a large investment, but in the long-term as you are reducing your disease burden. . .*" which indicated the direction of this relationship (negative correlation between farming practices and disease burden). Another participant then added the evidence to back it up by saying that "*the studies that are complete. . .. In the Netherlands and in Denmark, are on exactly that*". Later in the conversation a participant mentioned that it is "*fairly obvious around the good farming practices, and anything that we can do to improve the way we raise the food producing animals and keep them being in healthier conditions*", which provided an insight into the strength of the relationship and the level of evidence using language such as "*fairly obvious*". Overall, through the conversation between the participants, we were able to decipher that this was likely a strong, negative relationship, and that there was likely scientific evidence, in addition to personal and professional knowledge to back up this statement.

## Discussion

Overall, the participants spoke about the issue of AMR by combining their personal backgrounds and experiences with professional expertise, knowledge, and opinions from within their field and related fields. Through the sharing of expert knowledge, the participants were able to provide valuable quantifiable data about the current states of the nodes and the strengths and directions of the relationships.

### Key findings

It was noticed that most participants' comments were coded with the level "high" and "low", but not "medium". Similarly, "strong" relationship claims were much more apparent than

"weak" relationships. It is human nature to better remember the extremes rather than the average [45, 46]. This could explain why participants mainly focused on strong relationships; One would first think of those relationships that have large impacts or are known to be major drivers than those weaker relationships that may be less important. Participants may even feel that these weaker relationships are not worth mentioning as they are so far removed from the main issue. However, it is important to include all relationships, even if deemed small or insignificant, because they could become a large driver if another part of the system were to be changed or removed (e.g., purchasing of AMs on the black market is very limited in Sweden currently but could become more apparent if AMU is further limited through regulations, if the need for antimicrobials does not change).

We noticed that many statements referring to both the nodes and relationships were made based on tacit knowledge (e.g., personal, or professional knowledge and experience). As this was a secondary analysis and was not defined *a priori* as a major objective of the workshops, we did not probe the participants for the sources of their knowledge or the basis for their claims. The data collected were based on organic conversation. Thus, the nodes or relationships categorized as opinion or professional evidence may also have scientific evidence to support them that was not mentioned in the context of the workshop, and which should be verified with further expert engagement.

Finally, although Sweden has many of regulations around AMU and animal welfare [47], and they have low levels of AMR and AMU compared to many other countries [21, 48, 49], it is also possible that the participants were framing their claims to place Sweden in the best light possible, either consciously or unconsciously, to highlight their achievements to our Canadian research team and to those participants from outside of Sweden or other Nordic countries. Participants spoke very highly of Sweden in terms of their levels of factors such as regulations, disease, AMU, and resistance, and usually did so by comparing these to other countries (e.g., comparing another country's use of chlorine washes to the more preventative biosecurity practices in Sweden). However, the participants may not be aware of how large of an impact other practices, such as the heavy reliance on imported meat and fish, can have on the presence and exposure to AMR in Sweden. Therefore, future studies should cross-check the statements of the participants against available data (if it exists) to be able to confirm the claims of the participants.

## Limitations

This workshop took place in a specific setting (Stockholm Resilience Centre, Stockholm University, Sweden), at a specific time (Fall of 2019), with distinct participants from a variety of backgrounds related to AMR and the broader food system, and therefore cannot be generalized beyond the scope of this study. First, the perspectives of the participants present in the workshops shaped the direction of the conversation, and the outcomes could have varied if participants of other areas or backgrounds were in attendance. Furthermore, if this workshop were to be done during or after the COVID-19 pandemic, the findings may have been quite different. The pandemic could have out-competed AMR for importance and diminished the importance of certain nodes or relationships or changed the experts' views or estimates on certain aspects of the system (e.g., perceived changes in the levels of factors and relationships such as trust in science or leaders, socio-economic status of the population, rates of illness, or amount of AMU). Therefore, these findings are context specific and are limited to this time and place but were valid at the time of creation. However, there is no systems model that can be considered "correct" due to the everchanging landscape and therefore the CLD and the information described to populate the model represent an accurate estimation of the system

based on the participants' perspectives at the time [50]. Further studies in multiple different contexts (both in high income and low- and middle-income country contexts) can further expand our knowledge within each context and allow for comparisons between contexts to assess the generalizability of these findings [51].

Also, while there was an eclectic group of participants with varying levels of expertise in AMR and experience working with organizations across the European Union, Europe and internationally, future research would benefit to engage an even broader set of perspectives that were not represented in the workshops (e.g., the environment sectors) and further studies with an explicit focus on describing the values in quantifiable terms could help to identify more invaluable knowledge to further AMR knowledge from a variety of perspectives.

A common limitation associated with qualitative research is that the interpretation of the participants words, the coding, and the analysis and presentation of findings are subject to the researchers' own personal biases and intended outcomes [52, 53]. However, multiple processes were conducted to ensure the trustworthiness of our data [53]. For example, through discussion with members of the research team, we conducted inter-coder reliability and refinement of the analysis [52], as well as through triangulation [54] with other sources of data (results from a literature scan, [21, 55]), the potential biases associated with personal interpretation have been minimized to the best of our ability.

This study was also undertaken with a specific goal in mind (identify semi-quantitative data pertaining to the nodes and relationships). Therefore, there was a pre-conceived goal that may have limited the scope of what was identified in this analysis. However, using the stricter approaches found in framework analysis [37–39] permitted the identification and organization of the findings for use in future studies (e.g., model building) and streamlined the approach for use in mixed-methods research more broadly.

The final limitation of this study is that this study was a secondary analysis and thus is subject to the focus and limitations of how the primary workshop was undertaken. Key limitations related to the fact that participants were not always explicitly asked to discuss: the nodes and relationships in terms of semi-quantitative indicators such as the strengths of the relationships, the relative importance of drivers, or the type of evidence being used to inform their claims. However, without prompt, the participants provided great insight into many of the nodes and some relationships, but future studies could explicitly use participant input to provide quantitative estimates for the nodes and relationships through the use of participatory modelling approaches such as fuzzy cognitive mapping [56].

## Implications

Despite these limitations, this research has highlighted that qualitative data can be used to better understand complex One Health issues. Although it did not highlight any novel transmission pathways, through the engagement of multiple participants from a variety of backgrounds, it was possible to provide estimates to begin to quantify a One Health model of the system of drivers of AMR for areas of the system that have not yet been quantified or have limited data. Participatory modelling approaches have been used to identify and map out the major socioecological drivers of AMR in Europe [9], South-East Asia [57], New Zealand [58], and Tanzania [59], but these studies did not aim to estimate associated values. Qualitative methods have also been used to better understand the motivations that drive AMU in humans [60, 61], companion animals [62], and agriculture [63–67], and the drivers of prescribers in these settings [64, 68, 69] in both high-income (e.g., Denmark, United Kingdom) and low- and middle-income settings (e.g., Bangladesh, Thailand). These studies have helped to enrich the understanding of many drivers of AMU and AMR in these contexts which can help to inform the structure of the

system (e.g., identify nodes and relationships), however they do not provide quantifiable data to be utilized for modelling. Some studies have started to "quantify" factors and relationships using expert knowledge and input from the general public [70, 71]. A study in Switzerland engaged experts and consumers to discuss the relative importance of the multiple pathways in which humans can be exposed to AMR [70]. Similarly, in the United Kingdom, experts related to the companion animal veterinary field ranked the veterinary behaviours which contribute to AMR in veterinary practices [71]. These studies not only describe factors that may drive AMR but also provide estimates (rankings) of the relative importance of the factors which could provide a basis to begin to quantify these factors and relationships. These studies however are still limited in scope (human- or animal-centred) and fail to account for the complex socio-ecological drivers at a One Health scale. This study, however, include drivers from a wider range of the system (e.g., trade and economics, political and societal factors) and provides semi-quantitative estimates to factors and relationships that cross sectors and ecological scales.

Current quantitative dynamic models of AMR are limited in scope, both in terms of the populations captured but also in terms of the factors that are included [21–23]. These models typically include populations from one sector (e.g., humans or animals), in small settings (e.g., in a single hospital or farm), and basic factors such as AMU, hygiene practices [21–23]. Major reasons for the limited scope are: limited data availability, the lack of understanding of the relationships between sectors, and the limited ability to quantify relationships between sectors [22, 23]. The ability to estimate relationships and model potential outcomes of public health issues such as AMR is of great importance. However, with complex problems (such as with AMR) there are many drivers at play with complex nuances, such as socio-economic and cultural factors, that can drive human behaviour in unpredictable ways, making it difficult to quantify and model with current quantitative epidemiological methods. New methods to quantify these relationships, such as source attribution (e.g., risk assessment [72], metagenomics [73, 74], and whole genome sequencing [74, 75]), are being developed and tested but this work is still in its infancy. Furthermore, many of the drivers of AMR, such as how a person's understanding and awareness of AMR impacts their health-seeking behaviour or their food decisions, are abstract and difficult to quantify with traditional methods. Thus, from a disease modelling perspective, the engagement of experts to provide estimates into the current states of the nodes and the strength and direction of the correlations captured by the relationships to fill these gaps in data is an important intermediate step in the modelling of complex systems. For example, this model was able to highlight that the Swedish population has a high level of compliance and trust in political leaders, who are strongly influenced by science and academia for their decision-making. Thus, it can be assumed that if new scientific evidence were to come to light, it could largely impact policy and would be likely to be adopted by the general population. Alternatively, there is a large aversion to genetically modified foods and a high demand for organic and locally grown foods. Therefore, it is important to identify the strength of these pathways and other related pathways to be able to understand how to intervene. For example, if scientific evidence was identified that genetically modified foods were the only thing able to combat AMR and the government created policies that allowed and advocated for these foods in the markets, would this pathway be strong enough to alter the current relationship between the population demand for novel foods versus organic foods? These nuanced interactions are important to understand and quantify and would be of great use in models for policymaking.

## Conclusion

A workshop that included traditional and non-traditional experts in AMR provided valuable quantifiable data for the major drivers and interconnections related to AMR from both tacit

and explicit knowledge. This study helped us better understand the past, current, and potential future states of the factors that may influence AMR in the European One Health system. This study highlighted that the use of qualitative methods allowed us to better understand the issue of AMR and although these results are limited to this specific context, this study provided a strong knowledge basis of knowledge that could be used to help parameterize future models of this complex system.

## Acknowledgments

We would like to thank Sara Abdelrahman for input for the inter-coder reliability and Stephan Harbarth for contributions to the broader research project under the AMResilience team.

## Author Contributions

**Conceptualization:** Melanie Cousins, E. Jane Parmley, Amy L. Greer, Elena Neiterman, Irene A. Lambraki, Shannon E. Majowicz.

**Data curation:** Melanie Cousins.

**Formal analysis:** Melanie Cousins.

**Funding acquisition:** Didier Wernli, Peter Søgaard Jørgensen, Shannon E. Majowicz.

**Investigation:** Melanie Cousins.

**Methodology:** Melanie Cousins.

**Supervision:** Shannon E. Majowicz.

**Validation:** Elena Neiterman.

**Visualization:** Melanie Cousins.

**Writing – original draft:** Melanie Cousins.

**Writing – review & editing:** E. Jane Parmley, Amy L. Greer, Elena Neiterman, Irene A. Lambraki, Tiscar Graells, Anaïs Léger, Patrik J. G. Henriksson, Max Troell, Didier Wernli, Peter Søgaard Jørgensen, Carolee A. Carson, Shannon E. Majowicz.

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
