## [Decision Letter · Decision Letter 0]

20 Apr 2023

PONE-D-23-05382Is scientific evidence enough? Using expert opinion to parametrize a simulation model of antimicrobial resistance in Sweden.PLOS ONE

Dear Dr. Cousins,

Thank you for submitting your manuscript to PLOS ONE. After careful consideration, we feel that it has merit but does not fully meet PLOS ONE’s publication criteria as it currently stands. Therefore, we invite you to submit a revised version of the manuscript that addresses the points raised during the review process.

We look forward to receiving your revised manuscript.

Kind regards,

Jerome Nyhalah Dinga, PhD

Academic Editor

PLOS ONE

Journal Requirements:

3. We noted in your submission details that a portion of your manuscript may have been presented or published elsewhere. [While this manuscript has not been published elsewhere, an earlier version of this paper formed part of my PhD thesis, which I self-published on the University of Waterloo’s thesis repository (as required by my degree), http://hdl.handle.net/10012/18478. The manuscript submitted here is a revised version, that is different because of editorial changes based on co-authors recommendations, anonymization of country names to protect reputation, and changes of quotations displayed in the text to better depict examples. Please note I am the sole copyright holder of my thesis, and can assign copyright to the journal for this work if required] Please clarify whether this publication was peer-reviewed and formally published. If this work was previously peer-reviewed and published, in the cover letter please provide the reason that this work does not constitute dual publication and should be included in the current manuscript.

Additional Editor Comments:

Dear Authors,

At the current state, your manuscript does not warrant publication and you would be required to address the comments raised by reviewer(s) for further assessment.

Kindly do so and resubmit the manuscript for further assessment and decision.

Reviewers' comments:

Reviewer's Responses to Questions

**Comments to the Author**

1. Is the manuscript technically sound, and do the data support the conclusions?

Reviewer #1: Partly

2. Has the statistical analysis been performed appropriately and rigorously? 

Reviewer #1: N/A

3. Have the authors made all data underlying the findings in their manuscript fully available?

Reviewer #1: Yes

4. Is the manuscript presented in an intelligible fashion and written in standard English?

Reviewer #1: Yes

5. Review Comments to the Author

Reviewer #1: Importance of study topic

There is an urgent need for a quantitative and predictive approach to the AMR problem at a global level and in a One Health context. Ideally, such an approach would combine Big Data and mathematical modeling or machine learning to identify and exploit information regarding key drivers of AMR. Moreover, an ideal approach would encompass several hierarchical levels from microbial genomes and communities to hospital epidemics and national and global human infrastructures. Various intermediate approaches serve to facilitate this ultimate goal. The study at hand is one such intermediate approach, using expert opinion to identify key drivers of AMR. The outcome of an expert workshop was used to create a concept map of key drivers of AMR and links between them, an example of participatory modeling. As the authors themselves declare, identifying key drivers based on expert opinion is not ideal but is useful due to gaps in quantitative data. The approach the authors use is based on coding qualitative information into data regarding the importance (positive or negative effect and strength of effect) of particular factors in AMR as well as metadata such as the source of that information (e.g. professional experience or general knowledge). The result is a concept map with lines connecting different drivers.

Major issues:

The abstract and introduction are slightly misleading since they use expressions such as “parametrizing a model” that give the impression that a mathematical model of AMR drivers was developed. In fact, the outcome of the study is simply a concept map from an expert workshop. This could be stated more explicitly. The number of study participants is also only 17 and highly under-representative, mostly consisting of AMR professionals/experts from Sweden. Moreover, there was only 61 % consistency in the coding when three independent researchers coded the data. Even though the definition of the categories of the codes was refined and this was stated to result in consensus among the reviewers, it appears that intercoder reliability was not reassessed. Therefore, there may be considerable reliability issues caused by the subjective nature of the coding. In summary, the study makes a relatively weak and minor contribution to the understanding of AMR drivers. Therefore, to justify its consideration as a novel and rigorously conducted study to be accepted for publication in a scientific journal, the manuscript should be edited to (more) clearly answer the following questions:

(1) How is this study precisely distinct from and what is its novelty compared to Lambraki et al., 2022, that it is claimed to extend?

(2) How do the study findings expand on and relate to previous scientific discussion and findings concerning the drivers of AMR? Most of the drivers identified do not seem novel to me, and the results and discussion do not clearly contextualize the study findings in terms of previous research on AMR drivers.

In addition, another major point is that the authors should more transparently express that what they present in this study is (simply) a workshop concept map. The current use of elaborate language (e.g. model parametrization) can mislead readers regarding the methodology and importance of the study, considering it to be a mathematical modeling paper.

Minor comments:

The study is well written and structured.

6. PLOS authors have the option to publish the peer review history of their article (what does this mean?). If published, this will include your full peer review and any attached files.

Reviewer #1: No

---

## [Author Response · Author response to Decision Letter 0]

29 Jun 2023

We have addressed all reviewer comments in detail below. Line numbers below refer to the marked version of the revised manuscript. 

Editor comment: 

Response:

Editor comment:

Please provide additional details regarding participant consent. In the ethics statement in the Methods and online submission information, please ensure that you have specified what type you obtained (for instance, written or verbal, and if verbal, how it was documented and witnessed). If your study included minors, state whether you obtained consent from parents or guardians. If the need for consent was waived by the ethics committee, please include this information.

Response:

Per your editorial comments, we have added details regarding participant consent, including the specified type of consent (written) that was collected and a full ethics statement to the online submission information. 

Editor comment: 

We noted in your submission details that a portion of your manuscript may have been presented or published elsewhere. [While this manuscript has not been published elsewhere, an earlier version of this paper formed part of my PhD thesis, which I self-published on the University of Waterloo’s thesis repository (as required by my degree), http://hdl.handle.net/10012/18478. The manuscript submitted here is a revised version, that is different because of editorial changes based on co-authors recommendations, anonymization of country names to protect reputation, and changes of quotations displayed in the text to better depict examples. Please note I am the sole copyright holder of my thesis, and can assign copyright to the journal for this work if required] Please clarify whether this publication was peer-reviewed and formally published. If this work was previously peer-reviewed and published, in the cover letter please provide the reason that this work does not constitute dual publication and should be included in the current manuscript.

Response:

Regarding any previous publication of the work, please note that the only previous version of this work is my PhD thesis. This document is housed in my institution’s repository (http://hdl.handle.net/10012/18478) and I own the full copyright to the thesis. This document was not peer-reviewed nor is it formally published. The content of this manuscript is a revised version of the previous thesis version that was revised by a wider range of authors, the country names were de-identified within the quotations by the experts, and the results were further synthesized and framed within the broader context of current AMR research. 

Editor comment: 

Please include your full ethics statement in the ‘Methods’ section of your manuscript file. In your statement, please include the full name of the IRB or ethics committee who approved or waived your study, as well as whether or not you obtained informed written or verbal consent. If consent was waived for your study, please include this information in your statement as well.

Response: 

Per your editorial comments, we have added details regarding participant consent, including the specified type of consent (written) that was collected and a full ethics statement to the Methods Section (Lines 122-124). 

Editor comment:

Please include captions for your Supporting Information files at the end of your manuscript, and update any in-text citations to match accordingly. Please see our Supporting Information guidelines for more information: http://journals.plos.org/plosone/s/supporting-information.

Response:

We have ensured the manuscript meets PLOS ONE’s style requirements, including the reference style, title page format, sub-title formatting, and file names. Please note there are no supplementary materials to accompany this manuscript and therefore no captions are required. The supplementary materials that were referenced in the original manuscript were outlining where to locate specific figures within a different study (Lambraki et al., 2022). Therefore, to avoid confusion, these details have been removed and the references to the general article have remained.

Reviewer comment: 

The abstract and introduction are slightly misleading since they use expressions such as “parametrizing a model” that give the impression that a mathematical model of AMR drivers was developed. In fact, the outcome of the study is simply a concept map from an expert workshop. This could be stated more explicitly. 

Response: 

Thank you for this comment. We have revised the manuscript to outline our objectives more clearly with the results of the paper. Specifically, the title, abstract (Lines 4-10), rationale (Lines 67-80), and objectives (Lines 80-83) have been refined to better describe the intended goals of this manuscript. The title of the manuscript was updated to: “Is scientific evidence enough? Using expert opinion to fill gaps in data in antimicrobial resistance research” to better reflect the main research objective, which was to derive quantitative estimates from experts’ statements to begin to fill these data gaps. 

Please note that the causal loop diagram (CLD; the concept map to which you refer) was created during the original workshop and described/published by Lambraki et al., 2022. Specifically, Lambraki et al produced a visual image of the drivers and their interconnections. This manuscript advances the work of Lambraki et al by identifying quantitative estimates for the various drivers and their associations (e.g., strengths and directions) and adding them to the Lambraki et al CLD. The major contribution of this work was not to identify new concepts or divers of AMR, but as an intermediate step to fill gaps for where quantitative data does not yet exist or is limited. 

Reviewer comment: 

The number of study participants is also only 17 and highly under-representative, mostly consisting of AMR professionals/experts from Sweden. 

Response: 

To address the concerns about the under-representativeness of the study, we have added descriptions about participant characteristics (Lines 163-175) and information within the limitation sections (Lines 597-602). 

Overall, our participants had varying levels of expertise in AMR. About half were not experts in AMR, but rather were invited to participate because they possessed expertise in sectors or areas that may impact or be impacted by AMR. This was important to ensure our participants brought expertise from multiple sectors including those not traditionally considered when assessing upstream drivers and downstream impacts, including how those factors influence other factors and how they increase/decrease when impacted or impacting other factors.

We set out to recruit 12 to 20 participants representing diverse perspectives and successfully recruited 17 participants across human and animal sectors. Smaller sample sizes are common in qualitative research because the focus is on selecting information rich cases to adequately explore a phenomenon of interest, like AMR, in greater depth (Sandelowsk, 1996; Patton 2002; Cresswell & Planko Clark, 2011). Lincoln and Guba (1985) proposed that sample size determination be guided by the criterion of informational redundancy, that is, sampling stops when no new information is elicited. To ensure informational redundancy (saturation of issues discussed) our workshops were designed to provide participants with sufficient time to describe the factors/drivers and values that may inform the development of a quantitative mode, ensured participants had time to question, contest, and build on each other’s comments (facilitation techniques) and kept questioning and probing participants until we heard no new information emerge and participants themselves indicated to us they had nothing new to share. Moreover, our two workshop sessions each had different people participate (7 on day 1; 10 on day 2). Issues raised during the two sessions were similar, providing another indication of information redundancy. In addition, participatory systems dynamics modelling recognizes that while no systems model is 100% correct, because the external landscape is ever-changing, the CLDs and what participants described reflect the best estimation of the dynamics that influence AMR in the European context based on the perspectives of the participants at the time of data collection (Hummelbrunner, 2009). Thus, our approach to sampling followed accepted approaches to sampling in qualitative research and in establishing saturation of core issues/themes. Finally, this study provided semi-quantitative data for 83 nodes and various relationships and highlights that such data can be derived from this type of expert engagement. Because this study showed that useful information can be derived through these qualitative methods, future studies should expand participant engagement, including to those from sectors not at our workshop (e.g., environment) to populate the nodes and relationships of the CLD.

Reviewer comment: 

Moreover, there was only 61 % consistency in the coding when three independent researchers coded the data. Even though the definition of the categories of the codes was refined and this was stated to result in consensus among the reviewers, it appears that intercoder reliability was not reassessed. Therefore, there may be considerable reliability issues caused by the subjective nature of the coding. 

Response: 

Thank you for this comment. We would like to clarify that there was only 61% consistency between the reviewers in the original test coding. At this stage, the inconsistency came from one coder and the contradictions were minor, with most discrepancies due to the coder’s misunderstanding on the different sources of knowledge (mainly personal versus professional). We did not re-quantify intercoder reliability because the pre-agreed intent of our consensus-based approach to intercoder reliability was focused on fostering dialogue and reflexivity and surfacing discrepancies in coding to clarify, discuss and come to agreement on how to resolve the conflicting interpretations responsible, and using that information to refine the coding frame (e.g., better define codes that may be ambiguous) than a focus on a final score; reaching 100% consensus about our discrepancies was our indicator that satisfactory reliability had been reached and is an approach that is commonly used by others who conduct intercoder reliability checks with or without generating a score (Barbour, 2001; O’Connor & Joffe, 2022; Joffe & Yardley, 2003; Campbell et al., 2013; Cheunga & Tai, 2021). 

Details about the inter-coder reliability process has been added to the manuscript (Lines 252-258). 

Reviewer comment: 

In summary, the study makes a relatively weak and minor contribution to the understanding of AMR drivers. Therefore, to justify its consideration as a novel and rigorously conducted study to be accepted for publication in a scientific journal, the manuscript should be edited to (more) clearly answer the following questions:

1) How is this study precisely distinct from and what is its novelty compared to Lambraki et al., 2022, that it is claimed to extend?

Response: 

In response to the question posed, the following description was added to the methods section to explicitly outline how this study differs from the original workshop paper outlined by Lambraki et al., 2022: 

“While the original study outlined in Lambraki et al., 2022 produced a CLD of the system that drives AMR and thematically described the major areas of the system and potential places to intervene, it did not quantify the current state of the factors or put strengths to the relationships between the factors. Therefore, the transcripts from the workshop were re-analyzed to begin to quantify the CLD by identifying quantifiable data from the experts’ statements for the factors and strengths and directions of the relationships between the factors.” (Lines 113-119)

2) How do the study findings expand on and relate to previous scientific discussion and findings concerning the drivers of AMR? Most of the drivers identified do not seem novel to me, and the results and discussion do not clearly contextualize the study findings in terms of previous research on AMR drivers.

Response: 

As a response to the question above, the following clarifications were added to the Discussion section (Lines 636-663, 704-716). 

In short, the aim of this study was not to identify drivers (as was done in Lambraki et al, 2022) but to provide quantitative estimates to the drivers and the relationships between the drivers which have not or are hard to be quantified. Many studies of the drivers of AMR outline the system qualitatively but do not take the next step to begin to get experts to quantify the drivers. Many models of AMR are limited in scope because the estimates for cross-sector impacts have not been quantified (e.g., animal-environment or environment-human). Source attribution and whole genome sequencing work has begun to better understand these connections, but the data is still limited. Using expert knowledge in the interim is a good solution. 

Reviewer comment: 

In addition, another major point is that the authors should more transparently express that what they present in this study is (simply) a workshop concept map. The current use of elaborate language (e.g. model parametrization) can mislead readers regarding the methodology and importance of the study, considering it to be a mathematical modeling paper.

Response: 

Overall, we believe that the changes we have made above (reframing the title and objectives) have adequately addressed this concern. Furthermore, since we did not want to lose the concept of the future application of this method to further modelling of AMR, this has been touched on as a recommendation for future research in the Discussion section (Lines 701-704, 752-754). 

 

REFERENCES:

Barbour, R. S. (2001). Checklists for improving rigour in qualitative research: A case of the tail wagging the dog? British Medical Journal, 322, 1115–1117. https://doi.org/10.1136/bmj.322.7294. 1115

Campbell, J. L., Quincy, C., Osserman, J., & Pedersen, O. K. (2013). Coding in-depth semistructured interviews: Problems of unitization and intercoder reliability and agreement. Sociological Methods & Research, 42, 294–320. https://doi.org/10.1177/0049124113 500475

Joffe, H., & Yardley, L. (2003). Content and thematic analysis. In D. F. Marks & L. Yardley (Eds.), Research methods for clinical and health psychology (pp. 56–68). Sage.

Ka Ching Cheunga and Tai (2021). The use of intercoder reliability in qualitative interview data analysis in science education. https://doi.org/10.1080/02635143.2021.1993179

Lambraki IA, Cousins M, Graells T, Leger A, Henriksson P, Harbarth S, et al. (2022). Factors influencing antimicrobial resistance in the European food system and potential leverage points for intervention: A participatory, One Health study. PLoS One 17:1–19. https://doi.org/10.1371/journal.pone.0263914.

Lincoln YS, Guba EG. Naturalistic inquiry. London: Sage; 1985.

O’Connor and Joffee 2020: International Journal of Qualitative Methods; Volume 19: 1–13 DOI: 10.1177/1609406919899220

Patton MQ. Qualitative research and evaluation methods. 3rd Sage Publications; Thousand Oaks, CA: 2002.

Sandelowski M. One is the liveliest number: the case orientation of qualitative research. Res Nurs Health. 1996;19(6):525–9.

Williams B, Hummelbrunner R. Systems concepts in action. A practitioner's toolkit. Stanford, California: Stanford University Press; 2009

---

## [Decision Letter · Decision Letter 1]

9 Aug 2023

Is scientific evidence enough? Using expert opinion to fill gaps in data in antimicrobial resistance research.

PONE-D-23-05382R1

Dear Dr. Cousins,

We’re pleased to inform you that your manuscript has been judged scientifically suitable for publication and will be formally accepted for publication once it meets all outstanding technical requirements.

Kind regards,

Jerome Nyhalah Dinga, PhD

Academic Editor

PLOS ONE

Additional Editor Comments (optional):

Reviewers' comments:

Reviewer's Responses to Questions

**Comments to the Author**

1. If the authors have adequately addressed your comments raised in a previous round of review and you feel that this manuscript is now acceptable for publication, you may indicate that here to bypass the “Comments to the Author” section, enter your conflict of interest statement in the “Confidential to Editor” section, and submit your "Accept" recommendation.

Reviewer #1: All comments have been addressed

2. Is the manuscript technically sound, and do the data support the conclusions?

Reviewer #1: Yes

3. Has the statistical analysis been performed appropriately and rigorously? 

Reviewer #1: N/A

4. Have the authors made all data underlying the findings in their manuscript fully available?

Reviewer #1: Yes

5. Is the manuscript presented in an intelligible fashion and written in standard English?

Reviewer #1: Yes

6. Review Comments to the Author

Reviewer #1: Thank you for the revision. You have satisfactorily responded to the issues raised in my first review of the manuscript.

7. PLOS authors have the option to publish the peer review history of their article (what does this mean?). If published, this will include your full peer review and any attached files.

Reviewer #1: No

---

## [Editor Report · Acceptance letter]

11 Aug 2023

PONE-D-23-05382R1 

Is scientific evidence enough? Using expert opinion to fill gaps in data in antimicrobial resistance research. 

Dear Dr. Cousins:

I'm pleased to inform you that your manuscript has been deemed suitable for publication in PLOS ONE. Congratulations! Your manuscript is now with our production department. 

Kind regards, 

on behalf of

Dr. Jerome Nyhalah Dinga 

Academic Editor

PLOS ONE